# Safety and Efficacy of Hypofractionated Stereotactic Radiotherapy with Anlotinib Targeted Therapy for Glioblastoma at the First Recurrence: A Preliminary Report

**DOI:** 10.3390/brainsci12040471

**Published:** 2022-04-02

**Authors:** Yun Guan, Jing Li, Xiu Gong, Huaguang Zhu, Chao Li, Guanghai Mei, Xiaoxia Liu, Li Pan, Jiazhong Dai, Yang Wang, Enmin Wang, Ying Liu, Xin Wang

**Affiliations:** 1CyberKnife Center, Department of Neurosurgery, Huashan Hospital, Fudan University, 12 Wulumuqi Road (M), Shanghai 200040, China; yguan10@fudan.edu.cn (Y.G.); lijingck@fudan.edu.cn (J.L.); gongxiu2005@163.com (X.G.); zhuhg@fudan.edu.cn (H.Z.); lichao11@fudan.edu.cn (C.L.); meighai@126.com (G.M.); xiaoxia@fudan.edu.cn (X.L.); lipanmr@sina.com (L.P.); djzhadb@126.com (J.D.); janetcyj@163.com (Y.W.); wangem@fudan.edu.cn (E.W.); 2Neurosurgical Institute, Fudan University, 12 Wulumuqi Road (M), Shanghai 200040, China; 3National Center for Neurological Disorders, 12 Wulumuqi Road (M), Shanghai 200040, China; 4Shanghai Key Laboratory of Brain Function and Restoration and Neural Regeneration, 12 Wulumuqi Road (M), Shanghai 200040, China; 5Department of Pathology, School of Basic Medical Sciences, Fudan University, 138 Yi Xue Yuan Road, Shanghai 200032, China

**Keywords:** hypofractionated stereotactic radiotherapy, recurrent high-grade glioma, salvage treatment, anlotinib

## Abstract

(1) Background: Hypofractionated stereotactic radiotherapy (HSRT) and anti-vascular endothelial growth factor (VEGF) antibodies have been reported to have a promising survival benefit in recent studies. Anlotinib is a new oral VEGF receptor inhibitor. This report describes our experience using HSRT and anlotinib for recurrent glioblastoma (rGBM). (2) Methods: Between December 2019 and June 2020, rGBM patients were retrospectively analysed. Anlotinib was prescribed at 12 mg daily during HSRT. Adjuvant anlotinib was administered d1-14 every 3 weeks. The primary endpoint was the objective response rate (ORR). Secondary endpoints included overall survival (OS), progression-free survival (PFS) after salvage treatment, and toxicity. (3) Results: Five patients were enrolled. The prescribed dose was 25.0 Gy in 5 fractions. The median number of cycles of anlotinib was 21 (14–33). The ORR was 100%. Three (60%) patients had the best outcome of a partial response (PR), and 2 (40%) achieved a complete response (CR). One patient died of tumour progression at the last follow-up. Two patients had grade 2 hand-foot syndrome. (4) Conclusions: Salvage HSRT combined with anlotinib showed a favourable outcome and acceptable toxicity for rGBM. A prospective phase II study (NCT04197492) is ongoing to further investigate the regimen.

## 1. Introduction

Glioblastoma is the most frequently diagnosed malignant primary brain tumour in adults. Maximum surgical resection with six courses of temozolomide adjuvant chemoradiotherapy is the current standard of care in the first-line management of glioblastoma [1]. However, most patients still suffer from recurrence within eight months after primary treatment, and approximately 90% of recurrences occur within a 2 cm margin of the original tumour resection cavity [2]. The management of recurrent glioblastoma is highly challenging due to resistance to available therapeutic approaches, and treatment outcomes remain uniformly poor.

For recurrent glioblastoma (rGBM), several options have been studied, including surgery, re-irradiation, tumour-treating fields, and systemic therapy. Many second-line targeted agents and chemotherapy regimens have been examined in trials with limited success. The anti-vascular endothelial growth factor (VEGF) antibody bevacizumab has been demonstrated to prolong the progression-free survival (PFS) of glioblastoma (GBM); however, patients still progress after 3–6 months with an OS of 6–9 months after salvage treatments [3]. Anlotinib is a novel tyrosine kinase inhibitor that targets vascular endothelial growth factor receptor (VEGFR) 1/2/3, platelet-derived growth factor receptor (PDGFR), fibroblast growth factor receptors (FGFR) 1/2/3/4, c-Kit, and Ret. It has been reported to have a promising effect on tumour control in an rGBM case report [4]. However, as a salvage treatment, failure ultimately occurs. It is crucial to increase local treatment to reduce the risk of disease progression. The Radiation Therapy Oncology Group (RTOG) 1205 trial reported a prolonged PFS for bevacizumab with hypofractionated stereotactic radiotherapy (HSRT) compared with bevacizumab alone [5]. As the main pattern of failure remains local recurrence, it is crucial to optimize local control to improve survival.

Advances in stereotactic radiation can deliver high doses to tumours while limiting toxicity to normal structures. CyberKnife is a noncoplanar radiosurgery system that allows highly conformal image-guided radiotherapy and shows a promising tumour control effect for central nervous system tumours. A prior retrospective study at our centre showed the efficacy of hypofractionated stereotactic radiotherapy for recurrent high-grade glioma (rHGG) patients with mild toxicity. This study aimed to report the preliminary outcome of HSRT combined with anlotinib. To our knowledge, this is the first cohort of rGBM patients treated with HSRT combined with anlotinib.

## 2. Materials and Methods

### 2.1. Eligibility Criteria and Endpoints

This is a retrospective, cross-sectional study that was approved by the local ethics committee. Between December 2019 and June 2020, five rGBM patients received salvage HSRT with anlotinib at Huashan Hospital, Fudan University. All patients received surgery followed by standard chemoradiotherapy before recurrence. Recurrence was confirmed by the Response Assessment in Neuro-Oncology (RANO) criteria. Patients who were able to lie flat to receive radiotherapy and had Karnofsky Performance Status (KPS) scores higher than or equal to 70 were considered eligible for the regimen at our institution. All patients were treated at first recurrence within the radiation field and were not eligible for resection after neurosurgeons’ evaluation. Patients were informed that re-resection, re-irradiation, systemic therapy, and best supportive care were the treatment options and chose to receive the treatment after having fully understood and agreed to the potential harm and benefit.

The outcome endpoint was the objective response rate (ORR). Other endpoints included overall survival (OS) after HSRT, progression-free survival after salvage treatment, the best tumour response defined by the RANO criteria, and toxicity defined by the Common Terminology Criteria for Adverse Events (CTCAE) 5.0.

### 2.2. Baseline Evaluation and Treatment Delivery

Patients were immobilised with a custom thermoplastic mask and underwent localised 1.25-mm thin-slice computed tomography (CT, GE Light speed Ultra 16 Slice, San Francisco, CA, USA) and 2-mm thin-slice magnetic resonance imaging (MRI) including T1 post-contrast and T2 FLAIR images. CT and MRI scans were then fused using the planning system for contouring. HSRT was delivered by a CyberKnife Radiosurgery System (Accuray, Sunnyvale, CA, USA).

Radiation oncologists, neurosurgeons, and radiation physicists participated in tumour delineation and planning. The prescribed dose was 25.0 Gy in 5 fractions. The gross tumour volume (GTV) was defined as the gadolinium-enhanced tumour on the T1-weighted series. The clinical tumour volume (CTV) was considered equal to the GTV. The planning target volume (PTV) was a uniform 1-mm expansion of the CTV. Multiplan software was used for inverse planning. The prescribed isodose line to the PTV was determined according to the target volume, site, previous irradiation volume, and interval between treatments. Anlotinib (Tai-Tianqing Pharmaceutical Co., Ltd., Jiangsu, China) was prescribed at a dose of 12 mg daily for 14 consecutive days every 3 weeks from the first day of HSRT.

### 2.3. Assessment and Toxicity

All patients underwent a clinical and radiological follow-up every two months after HSRT. If there was any significant deterioration in the patient’s performance, an MRI was performed immediately. The radiological examination included MRI and other necessary examinations, such as MRI-based spectroscopy, perfusion MRI, and methionine positron emission tomography. The KPS after treatment, adverse event occurrence, and associated clinical outcomes were recorded. Toxicity was assessed using the CTCAE 5.0.

### 2.4. Statistics

The outcome measures considered were the objective response rate based on the proportion of patients with a best overall response of a confirmed complete response (CR) or partial response (PR). Other measures included overall survival after HSRT, defined as survival from the time of the completion of HSRT to death due to any cause, progression-free survival after salvage treatment, and treatment-related toxicities.

The CTCAE 5.0 was used to assess toxicity. The number of events, number of subjects, and incidence rate are used to describe the measurement. The maximum, minimum, and median values are used to describe the measurements of patient characteristics.

## 3. Results

### 3.1. Patient Characteristics

Five glioblastoma patients with clinical and radiographic evidence of recurrence were treated with HSRT between December 2019 and June 2020. All patients were initially treated with a maximum safe resection of gross total resection and adjuvant radiation treatment with a median dose of 60 Gy in 30 fractions with concurrent and maintenance temozolomide. The GTV of adjuvant radiation after surgery was defined by the post-contrast T1 and T2 fluid-attenuated inversion recovery sequences. GTV was expanded 1–2 cm to create CTV. PTV with a 3–5 mm margin was added to the CTV. All patients had information on methyl-guanine-methyltransferase (MGMT), isocitrate dehydrogenase 1 (IDH1), 1p/19q co-deletion, and telomerase reverse transcriptase (TERT) after initial resection. Five patients were TERT- and MGMT-positive, one patient had a 1p/19q co-deletion, and no patient was IDH1-positive. Three patients were male and two were female. The median age was 51 years (range 43–60 years). The KPS score at the time of salvage treatment ranged from 70 to 90. The median time from initial diagnosis to salvage HSRT was 10.4 months, with a range of 7.0 to 14.8 months. The median PTV was 26.9 cm^3^ (5.5–54.4 cm^3^). The treatment was delivered daily, and the dose was 25 Gy in five fractions with a median isodose line of 68% (65–70%). Patient characteristics are listed in Table 1.

### 3.2. Compliance and Toxicities

All patients received the planned radiation dose without interruption. The median number of cycles of anlotinib administered were 21 and ranged from 14 to 33 cycles. No acute clinical morbidity was observed. Grade 2 hand-foot syndrome was observed in two patients during cycles 8 and 10. Anlotinib was discontinued for one week in these two patients. The symptoms were relieved after dermatologic treatment, and the regimen was continued. Details are shown in Table 2. No operations or hospitalisation was required related to acute or delayed toxicity of HSRT and anlotinib.

### 3.3. Treatment Outcomes

The patients were assessed by the RANO criteria. Three (60%) patients had the best outcome of PR, and two (40%) achieved CR; the ORR was 100% (Figure 1A). The follow-up from the time of HSRT ranged from 20 to 26 months. By the end of the study, four patients had progressive disease (PD) and one patient died of tumour progression (Figure 1B). The overall survival rates following the salvage treatment were 100% and 80% at one and two years, and the PFS rates were 60% and 40%, respectively (see Appendix A).

## 4. Discussion

Recurrent glioblastoma has been reported to have a poor prognosis. Due to its therapeutic resistance and aggressiveness, its clinical management is challenging. GBM is a vascularised tumour that produces VEGF. Anti-VEGF treatments have been widely used in recurrent GBM. The mechanism of anti-VEGF treatments may have two aspects. First, inhibiting VEGF and its receptor reduces tumour angiogenesis to produce a hypoxic environment and inhibits tumour growth [6]. Second, the tumour vessel diameter was normalised, and the basement membrane was thin. A reduced volume of tumour microvessels has been reported to be related to longer survival [7].

Bevacizumab is approved for treating recurrent glioblastoma by the US Food and Drug Administration and has become a recommended treatment in the National Comprehensive Cancer Network (NCCN) guidelines, with several phase II and III randomised trials indicating a prolonged PFS compared with chemotherapy alone [8,9,10]. A phase III RCT reported a prolonged median PFS (4.2 vs. 1.5 months) in the bevacizumab and Lomustine groups compared with the Lomustine alone group. However, this trial did not find a difference in OS between the two groups. The grade 3 to 5 toxicity rate in the experimental group was 63.6% [10]. Friedman et al. reported a phase II randomised controlled trial (RCT) in which a higher 6-month PFS rate (50.3% vs. 42.6%) and better ORR (37.8% vs. 28.2%) were observed in the bevacizumab with irinotecan group than in the bevacizumab alone group [9]. Other anti-angiogenic drugs, including sorafenib, pazopanib, sunitinib, etc., were reported in phase I and II trials treating rGBM. The ORR reported for anti-VEGF treatments for rGBM ranged from 6% to 30% (Table 3), and the 6-month PFS ranged from 3% to 63%. The treatment-related toxicity was mild for these anti-VEGF treatments. However, the efficacy seems to be unsatisfactory.

Anlotinib is an oral novel multi-target tyrosine kinase inhibitor targeting the VEGF1/2/3 receptor, fibroblast growth factor receptor and platelet-derived growth factor receptor. It inhibits more targets than bevacizumab, sunitinib, sorafenib, etc. and has been reported to reduce both tumour proliferation and angiogenesis [3]. Lv et al. published the first case report of the administration of 12 mg anlotinib to an rGBM patient. The patient achieved a PR after 26 days, but the tumour progressed in two months [4]. Wang et al. reported a recurrent GBM patient with an FGFR-TACC3 fusion who was administered anlotinib 12 mg and temozolomide 100 mg/m^2^. The patient achieved a PR after two months and maintained stable disease for more than 17 months [11].

Several reports have suggested that re-irradiation has a reasonable efficacy with acceptable safety profiles in selected patients with recurrent GBM. However, for rGBM, salvage treatment failure ultimately occurs. It is crucial to increase local treatment to reduce recurrence risk. In a meta-analysis, a highly conformal technique with a hypofractionated regimen (e.g., 25 Gy in five fractions or 35 Gy in 10 fractions) is recommended, considering the volume and location of the recurrent tumour. The RTOG 1205 trial reported a prolonged PFS with anti-VEGF treatment with HSRT compared with bevacizumab alone [5]. Philip et al. theorized that additional anti-VEGF treatment sensitised the tumour endothelia to radiotherapy and induced apoptosis [12]. New-generation automated noncoplanar HSRT delivery systems can deliver high-dose treatment by limiting the dose to normal structures and can provide a higher local treatment intensity for recurrent tumours. In this study, the ORR rate of salvage treatment was 100% in two CR and three PR patients. The ORR was higher than other results of anti-VEGF treatments [13,14,15,16,17,18,19,20,21,22,23,24,25], which ranged from 6% to 30% (Table 3). There may be several possible reasons for the promising treatment outcomes. Patient selection may be a reason for good outcomes. All patients had a KPS of 70 or higher, and HSRT was performed after the first recurrence. Moreover, the administration of HSRT increased the local treatment intensity. The preliminary result of RTOG 1205 also reported an increased PFS in the intensified treatment groups. Additionally, patients with a smaller tumour volume may have a better response. The two CR patients (Figure 2A,B) in this study had a relatively smaller PTV (7.08 and 5.53 cm^3^) than the three PR patients (26.94, 44.33, and 54.41 cm^3^).

Salvage HSRT was administered with a full dose of 25 Gy/5 fx for all five patients without any interruption. No radiation necrosis occurred during the follow-up. Grade 2 hand-food syndrome was found in two patients (40%), and rash and hypertension were observed in one patient (20%). These adverse effects were considered to be related to anlotinib. In a phase II randomised trial of non-small-cell lung cancer patients, 28.33% of the subjects had grade 2 hand-foot syndrome, and grade 2 hypertension was observed in 55% of patients [26]. These toxicities were also observed in our study.

The study had some limitations due to its retrospective nature: an inherent patient selection bias was created when the physicians chose eligible patients to receive the regimen. The treatment option was provided for patients with high KPS scores who were not willing to receive standard intravenous bevacizumab treatment. Thus, the cohort was enriched with patients with a better prognosis. Another limitation was that recurrence before salvage treatment was diagnosed by radiological parameters according to the RANO criteria, which is a common practice [27]. However, the lack of biopsy samples limited the information on tumour genomic characterisations. It is crucial to consider whether the previously detected mutation still presents as the dominant clone at the time of recurrence [28]. Further investigation is warranted to explain the potential treatment mechanisms and select good responders to the regimen.

Despite the limitations, this study provides initial evidence of a promising outcome using salvage HSRT with anlotinib in a real-world scenario. Responses were observed in all rGBM patients included in the study. Further investigation is needed to identify patients who can benefit from this regimen. A prospective phase II study HSCK-002 (ClinicalTrials.gov identifier: NCT04197492) is ongoing to further investigate the value of HSRT with anlotinib.

## 5. Conclusions

Salvage radiosurgery with anlotinib appeared to achieve a clinical benefit with acceptable toxicity for rGBM patients in this preliminary report. A prospective phase II study (NCT04197492) is ongoing to further investigate the value of HSRT with anlotinib in rHGG.

## Figures and Tables

**Figure 1 brainsci-12-00471-f001:**
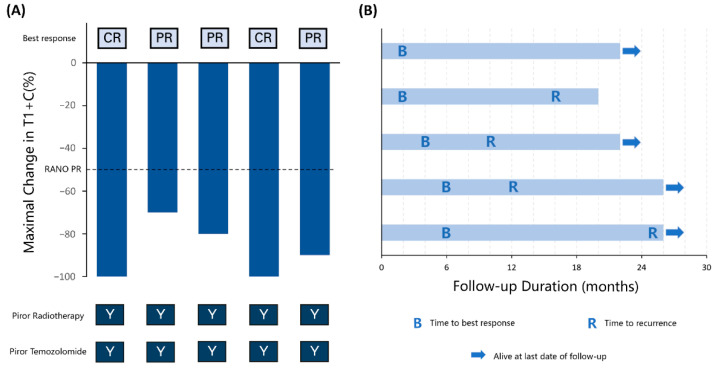
(**A**) Maximal change in the product of the perpendicular diameter in MRI T1 contrast before and after HSRT with anlotinib in each patient. CR, complete response. PR, partial response. RANO, Response Assessment in Neuro-Oncology. Y, yes. (**B**) Follow-up duration, time to the best response, and time to recurrence in each patient.

**Figure 2 brainsci-12-00471-f002:**
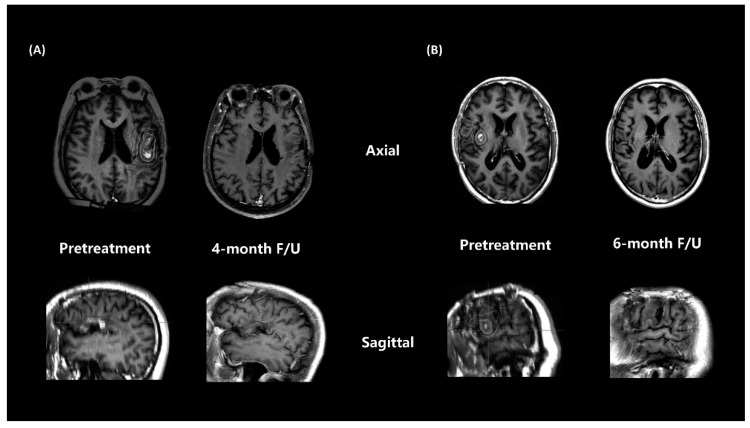
Contrast-enhanced MRI T1 of responses to HSRT and anlotinib, including (**A**) patient case 1 and (**B**) patient case 4, who achieved a complete response.

**Table 1 brainsci-12-00471-t001:** Patient characteristics and treatment outcomes.

Case	Age Sex	Interval between Initial Diagnosis and HSRT (Months)	Upfront RT Dose/fx	Upfront Chemotherapy (Cycles)	MGMT	IDH1	1p/19q
1	60 Male	12.6	60Gy/30	TMZ (12)	+	-	-
2	46 Female	10.4	60Gy/30	TMZ (6)	+	-	-
3	55 Female	14.8	60Gy/30	TMZ (12)	+	-	+
4	51 Male	10.0	60Gy/30	TMZ (4)	+	-	-
5	43 Male	7.0	60Gy/30	TMZ (4)	+	-	-
Case	TERT	Recurrent Lesion	Recurrent PTV (cm^3^)	KPS at HSRS	Dose (iso-dose line)	Cycles of Anlotinib	F/U Interval from HSRS (months)
1	+	Left Frontal Lobe	7.08	80	68	15	10
2	+	Left Frontal Lobe	26.94	80	65	14	10
3	+	Left Occipital Lobe	54.41	70	70	9	6
4	+	Right Frontal Lobe	5.53	90	70	4	4
5	+	Left and Right Frontal Lobe	44.33	90	68	8	6

**Table 2 brainsci-12-00471-t002:** Best treatment outcomes and adverse events that occurred in rGBM patients.

Outcomes/AE	Total No. of Patients	No. of Patients		
		Grade 1	Grade 2	Grade 3
ORR	5 (100%)	N/A		
CR	2			
PR	3			
Haematologic				
Thrombocytopenia	1	0	1	0
Nonhaematologic				
Hand foot syndrome	2	0	2	0
Rash	1	0	1	0
Hypertension	1	0	1	0

**Table 3 brainsci-12-00471-t003:** Reported anti-angiogenic treatment for recurrent glioblastoma.

Author, Year	Treatment	Phase (Sample Size)	Outcome (ORR Rate%)	Median PFS (Months)	Median OS (Months)	6-Month PFS
Reardon, 2018 [13]	Trebananib	II (11)	2CR (18)	0.7	11.4	N/A
Reardon, 2005 [14]	Imatinib	II (33)	3PR (9)	3.3	N/A	27.0%
Iwamoto, 2010 [15]	Pazopanib	II (35)	8PR (22)	3.0	8.1	3.0%
Pan, 2012 [16]	Sunitinib	II (16)	0	N/A	12.6	16.7%
Hutterer, 2014 [17]	Sunitinib	II (40)	0	2.0	9.2	12.5%
Hassler, 2014 [18]	Imatinib	II (24)	2PR (8)	3.0	6.2	N/A
Batchelor, 2010 [19]	Cediranib	II (131)	1CR, 17PR (14)	3.0	8.0	16.0%
Gerstner, 2015 [20]	Cediranib	I (45)	2CR, 2PR (9)	1.9	6.5	4.4%
Chheda, 2015 [21]	Vandetanib	I (19)	2PR (11)	1.9	7.2	63%
McNeill, 2014 [22]	Vandetanib	II (32)	2PR (6)	1.7	5.6	N/A
Duerinck, 2016 [23]	Axitinib	II (22)	2CR, 4PR (27)	N/A	6.7	34%
Lee, 2012 [24]	Sorafenib	I/II (18)	2PR (11)	1.8	N/A	N/A
Groot, 2020 [25]	Aflibercept	II (27)	8PR (30)	N/A	N/A	N/A

## Data Availability

Research data are stored in an institutional repository and will be shared upon request to the corresponding author.

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
