# Peer review of "Safety and Efficacy of Hypofractionated Stereotactic Radiotherapy with Anlotinib Targeted Therapy for Glioblastoma at the First Recurrence: A Preliminary Report"

_brainsci, 2022, doi:10.3390/brainsci12040471_

Round 1
Reviewer 1 Report
Dear author
Thank you for providing such an interesting article on the role of SRS + immunotherapy in the management of recurrent glioblastoma.
1. In the introduction, please mention that maximum surgical resection and adjuvant chemoradiation and six courses of adjuvant temozolomide is the current standard of care in the first line management of glioblastoma and cite this newly published paper.
Attarian F, Taghizadeh-Hesary F, Fanipakdel A, et al. A Systematic Review and Meta-Analysis on the Number of Adjuvant Temozolomide Cycles in Newly Diagnosed Glioblastoma. Frontiers in Oncology. 2021 ;11:779491. DOI: 10.3389/fonc.2021.779491. PMID: 34900732; PMCID: PMC8651479.
2. There are some grammatical errors that should be corrected. For example, in the title "the" is missing where you write "at (the) First Recurrence". Please read the text once more.
3. In the methods, please mention the type of study (i.e. cross sectional!?)
Reviewer 2 Report
Interesting work, nicely presented.
Still, limited data with only 5 patient and quite short follow up, an extension with more mature data would be nice after reaching recurrence. Even selection bias is a bit concerning.
KPS mentioned with range 60-90, at another point "KPS in all patients more than 70"
Worth the publication as preliminary result
Reviewer 3 Report
Despite the limitations, the study shows a new perspective for the treatment of patients with recurrent GBM. It is not possible to draw conclusions that will change clinical practice but it is evidently only the basis of an ongoing prospective study.
The manuscript shows the preliminary data of a study focused on the treatment of GBM relapses, a highly topical topic.
It highlights the low toxicity impact of the drug in association with radiotherapy for the treatment of a disease with a highly poor prognosis.
Unfortunately, the retrospective nature and the small sample of patients analyzed do not allow us to provide solid conclusions.
However, the paper is well written and the results are clearly exposed addressing the main question posed.
